# Identification, Characterization, and Antibacterial Evaluation of Five Endophytic Fungi from *Psychotria poeppigiana* Müll. Arg., an Amazon Plant

**DOI:** 10.3390/microorganisms12081590

**Published:** 2024-08-05

**Authors:** Sonia Mendieta-Brito, Mahmoud Sayed, Eunjung Son, Dong-Seon Kim, Marcelo Dávila, Sang-Hyun Pyo

**Affiliations:** 1Centro de Tecnología Agroindustrial, Universidad Mayor de San Simón, Cochabamba 00591, Bolivia; 2Division of Biotechnology, Department of Chemistry, Faculty of Engineering, Lund University, SE-22100 Lund, Sweden; 3Department of Botany and Microbiology, South Valley University, Qena 83523, Egypt; 4KM Science Research Division, Korea Institute of Oriental Medicine, 1672 Yuseong-Daero, Yuseong-Gu, Daejeon 34054, Republic of Korea

**Keywords:** endophytic fungi, *Psychotria poeppigiana* Müll. Arg., *Neopestalotiopsis* sp., *Penicillium* sp., *Aspergillus* sp., cultivation and molecular identification, antibacterial activity, chemical profiling and active group

## Abstract

Endophytic fungi, residing within plants without causing disease, are known for their ability to produce bioactive metabolites with diverse properties such as antibacterial, antioxidant, and antifungal activities, while also influencing plant defense mechanisms. In this study, five novel endophytic fungi species were isolated from the leaves of *Psychotria poeppigiana* Müll. Arg., a plant from the Rubiaceae family, collected in the tropical Amazon region of Bolivia. The endophytic fungi were identified as a *Neopestalotiopsis* sp., three *Penicillium* sp., and an *Aspergillus* sp. through 18S ribosomal RNA sequencing and NCBI-BLAST analysis. Chemical profiling revealed that their extracts obtained by ethyl acetate contained terpenes, flavonoids, and phenolic compounds. In a bioautography study, the terpenes showed high antimicrobial activity against *Escherichia coli*. Notably, extracts from the three *Penicillium* species exhibited potent antibacterial activity, with minimum inhibitory concentration (MIC) values ranging from 62.5 to 2000 µg/mL against all three pathogens: *Escherichia coli*, *Staphylococcus aureus*, and *Enterococcus faecalis* (both Gram-positive and Gram-negative bacteria). These findings highlight the potential of these endophytic fungi, especially *Penicillium* species as valuable sources of secondary metabolites with significant antibacterial activities, suggesting promising applications in medicine, pharmaceuticals, agriculture, and environmental technologies.

## 1. Introduction

Fungi have emerged as a rich source of bioactive and structurally unique secondary features [1,2]. Due to the extensive chemical and biological diversity of these secondary metabolites, exploring the fungal genus continues to offer opportunities for discovering novel lead compounds with potential for pharmaceutical drug development and biocontrol agent synthesis [3]. In recent years, endophytic fungi have garnered increasing attention in research circles, attributed to the identification of numerous novel compounds [4,5,6]. Endophytic fungi, microorganisms residing internally within various host plant tissues asymptomatically without inducing apparent disease [7,8,9], possess the ability to engage in complex interactions with their host plants. Through these interactions, plants can modulate endophytic metabolic processes to produce molecules with protective functions, benefiting both the microbe and host [10,11].

The cohabitation of endophytes within host plants may aid in their adaptation to biotic and abiotic stress factors [7,12]. Endophytic fungi have demonstrated the potential to produce bioactive metabolites that influence plant defense mechanisms, thereby enhancing the survival of both symbiotic entities. Generally, leaves exhibit a more diverse community of fungal endophytes compared to other plant parts [7,13]. Given their potential as an abundant source of new metabolites, crude extracts from these microorganisms hold promise as an alternative approach, with the capacity for industrial-scale production of bioactive compounds, thereby reducing final product costs and contributing to plant species conservation [14].

Consequently, this research explores the capability of endophytic fungi to generate novel secondary metabolites with diverse biological activities, such as antibacterial, antioxidant, and antifungal properties. Despite being recognized as promising sources of novel active compounds, the true potential of endophytic fungi, in terms of biological activity and biotechnological applications, remains largely unexplored [7], particularly in the context of tropical plant endophytes and their pharmacological potential [15]. The Amazon forest harbors the highest diversity of plant species on Earth, alongside an extraordinary abundance of fungi [16,17]. Research efforts have focused on collecting biological and ecological data on both rare and common fungal species to develop a deeper understanding of their roles in ecosystem functioning, particularly in light of predictions of global warming [17]. However, research on the identification and role of endophytic fungi remains underreported.

Meanwhile, Lidilhone Hamerski et al. conducted a comprehensive review exploring the natural product diversity and pharmacological properties of secondary metabolites produced by endophytic fungi associated with various genera of Rubiaceae [9]. Rubiaceae stands as the fourth largest angiosperm family, comprising approximately 617 genera and 13,000 species of herbs, shrubs, and trees, distributed worldwide, with a notable concentration in tropical and warm regions [9,18]. This botanical family showcases a remarkable diversity of chemical substances, including iridoids, anthraquinones, indole alkaloids, terpenoids, flavonoids, and alkaloids [9,18,19,20]. Several species within the Rubiaceae family have found extensive use in folk medicine, with evidence of anti-inflammatory, analgesic, antibacterial, mutagenic, antiviral, and antioxidant activities. One such species within Rubiaceae, *Psychotria poeppigiana* Müll. Arg. (now accepted as *Palicourea tomentosa* (Aubl.) Borhidi), is native to Central and South America, specifically Bolivia, Venezuela, and Brazil [21]. Traditionally, it has been employed in medicinal practices for the treatment of inflammation and pain. Additionally, *Cephaelis elata*, a synonym for *Psychotria poeppigiana*, has been utilized for addressing dementia [21]. Recent research has delved into the antioxidant, anti-inflammatory, and anti-acetylcholinesterase (AChE) activities of the chemical composition of *P. poeppigiana* essential oil extracted from its leaves (EOPP) [21]. However, investigations into the endophytic fungi associated with *P. poeppigiana*, particularly concerning their extract’s antibacterial activity, have not been performed.

With this context, in this study, *P. poeppigiana* Müll. Arg., a plant belonging to the Rubiaceae family, was collected from the tropical Amazon region of Cochabamba, Bolivia. From the leaves of this plant, five new species of endophytic fungi (*Neopestalotiopsis* sp. SMB-23, *Aspergillus* sp. SMB-27, and three species of *Penicillium* sp. SMB-24, SMB-25, and SMB-26 were isolated and identified using 18S ribosomal RNA sequences, analyzed through the online NCBI-BLAST tool. Phylogenetic trees were also constructed using the same tool. Subsequently, the resulting extracts underwent antibacterial testing to determine the minimal inhibitory concentration (MIC), revealing significant inhibition against pathogenic bacteria in the extracts from three *Penicillium* species. The chemical profiles of the extracts were investigated to identify the main groups of chemical compounds.

## 2. Materials and Methods

### 2.1. Materials

The materials utilized encompassed basic microbiology and chemistry supplies for the preparation and handling of culture media, bacterial strains, and extracts from endophytic fungal strains. Potato dextrose agar (PDA) and potato dextrose broth (PDB) were meticulously prepared in the laboratory following established protocols [22] or purchased from Merck (Darmstadt, Germany). The culture media for the bacterial strains, Nutrient medium and Brain Heart Infusion BHI, were purchased from MBcell, the Republic of Korea. Commercially procured Mueller–Hinton Agar, Trypto-Casein Soy Broth (TSB), and Trypto-Casein Soy Agar (TSA) were sourced from OXOID manufacturing. Additionally, the Quick-DNA Fungal/bacterial Miniprep Kit Zymo Research (Irvine, CA, USA) and the Gene JET PCR Purification Kit Thermo Scientific, (Waltham, MA, USA) were employed for DNA extraction and purification processes following the protocol introduced by the manufacturer. For polymerase chain reaction (PCR) amplification, specific primers (ITS3, ITS5, EF4f, and Fung5r) were utilized Integrated DNA Technologies IDT, (Coralville, IA, USA). Ethyl acetate, n-hexane, ethanol, sodium hypochlorite, *p*-anisaldehyde, ferric chloride, and aluminum chloride were acquired from Sigma-Aldrich (Saint Louis, MO, USA). TLC Silica gel 60 F254 chromatographic plates (aluminum sheets 20 × 20 cm, silica gel matrix, fluorescent indicator) and Mueller–Hinton (M-H agar) were purchased from Merck (Darmstadt, Germany).

### 2.2. Isolation of Endophytic Fungi from Psychotria poeppigiana Müll. Arg.

The collection of specimens of *P*. *poeppigiana* Müll. Arg., a plant species, was conducted in the Amazon region known as Valle del Sacta at an altitude of 240 m above sea level (17°05′12″ S and 64°46′19″ W), located within the Carrasco province of Cochabamba, Bolivia (Figure 1).

This plant, belonging to the Rubiaceae family, was identified using taxonomic keys of the Amazonian flora and the regional flora of the Sacta Valley, corroborated by reference to the “M. Cárdenas” National Forest Herbarium (BOLV). The collected plant parts (small branches with leaves) were stored in chilled containers at 4 °C for transportation to the laboratory [23,24].

Endophytic fungi were isolated from the aerial parts of the *P*. *poeppigiana* Müll. Arg. plant. Healthy leaves underwent surface disinfection through sequential washes with running tap water, 70% ethanol for 2 min, 1% hypochlorite for 1 min, and sterilized water for 2 min [23,24]. For the isolation of endophytic fungi, the plant material was cut into approximately 5 mm^2^ pieces and inoculated into Petri dishes (8–10 fragments per plate) containing (PDA) supplemented with 100 µg/mL chloramphenicol. The dishes were then incubated at 22 °C for 15 days. Mycelium was transferred to new plates, isolated, purified into pure strains, and stored in darkness. Depending on the culturable endophytes isolated, these were transferred to cryovial tubes containing 10% glycerol in PDB medium, cultured for 1 week at room temperature, and stored at −20 °C for later use in fermentations and biological tests [25,26]. The isolated fungi were deposited in the Microbiology laboratory of the Centro de Tecnología Agroindustrial (CTA), Universidad Mayor de San Simón Cochabamba, Bolivia.

### 2.3. Identification of Endophytic Fungi

Five fungi were isolated from the leaves of the *P. poeppigiana* Müll. Arg. plant and underwent identification using both classical taxonomies, based on morphological characteristics and molecular techniques. The macromorphological characteristics were analyzed after seven days of culturing at 30 °C in Petri dishes (10 mm × 100 mm) containing PDA. Macroscopic vegetative traits, including color, texture, topography, diffuse pigmentation, and colony border, were examined. Additionally, the back topography of the colony was assessed [27]. Microscopic morphology, encompassing hyphae and reproductive structures, was evaluated using the microculture technique in PDA for 5–7 days and YES for 7–10 days [28]. Stained macroscopic glass slides using Lactophenol blue solution (Merck, Stockholm, Sweden) were observed under an optical microscope (LRI-OLYMPUS-100×/0.65, Tokyo, Japan) [10,29], and the obtained results were compared with taxonomic keys [28,30,31].

Furthermore, molecular taxonomy was employed for identification using specific PCR primers to amplify the endophyte ITS1-5.8S-ITS2 region and 530 bp conserved region of the 18S rDNA gene. Fungal samples (50–100 mg wet weight) were obtained from fresh cultures grown in Petri dishes containing PDA for 7 to 15 days at a temperature between 28 and 30 °C [29,32,33]. Genomic DNA extraction from the fungi was performed using the Quick-DNA extraction kit Zymo Research (Irvine, CA, USA), following the manufacturer’s instructions, and stored at −20 °C in 100 µL DNA elution buffer.

The amplification of the ITS1-5.8S-ITS2 region and 530 bp conserved region of the 18S rDNA gene was conducted via polymerase chain reaction (PCR) using the primers listed in Table 1. The EF4f/Fung5r primers were used for SMB-24, 26 and 27, while the forward ITS5F/ITS3R primer was used for SMB-25. The PCR reactions were carried out in a T100 Thermal Cycler BIO-RAD Laboratories Inc., (Hercules, CA, USA) under specified conditions (Table 1).

The PCR products were visualized via agarose gel electrophoresis by running 2 µL of the PCR product in 1.0% (*w*/*v*) agarose gel with 19 tris/borate electrophoresis buffer (TBE). The gel was stained with GelGreen Nucleic Acid Gel Strain, 10,000× in water Fisher Scientific, (El Paso, TX, USA), and compared with a reference marker, GeneRuler 1 Kb DNA Ladder, Thermo Fisher Scientific (Waltham, MA, USA). The PCR products were purified using a PCR purification kit following the manufacturer protocol and were then subjected to direct sequencing using the same PCR primers, which were performed by Eurofins Scientific (Augsburg, Germany) [10,29].

Following sequencing, sequence analysis and alignment were performed using the online BLASTtool (blastn) in NCBI GenBank databases (website: www.ncbi.nlm.nih.gov/blast. accessed on 13 May 2024) to assess DNA similarities. The sequence of the endophytic fungal isolate was aligned with representative sequences of reference taxa, including members from other orders, obtained from GenBank. This alignment was crucial for resolving phylogenetic relationships.

To construct the phylogenetic trees, multiple sequence alignment and molecular phylogeny were conducted using Molecular Evolutionary Genetics Analysis (MEGA) 10.2.4 [29]. The phylogenetic trees were constructed from models generated by MEGA software (www.megasoftware.net, freeware, Accessed on 15 April 2024), with bootstrap analysis performed using 1000 replications to assess the reliability of the node tree [34]. Finally, the obtained sequences were deposited in GenBank for future reference and analyses.

### 2.4. Preparation and Chemical Profile of Extract from Endophytic Fungi

Five fragments of fungal mycelium (each measuring 5 × 5 mm^2^), from every strain of endophytic fungus under investigation, were excised from the PDA plates and inoculated into 500 mL Erlenmeyer flasks containing 200 mL of PDB liquid medium composed of potato broth (200 g/L) and dextrose (20 g/L) [22]. The cultures were incubated under shaking conditions at 200 rpm and 30 °C for 15 days, following the methodology outlined by Bose et al. [35], with certain modifications. To monitor biomass production under these culture conditions, the amount of biomass produced by each strain over the 15-day fermentation period was quantified. At the end of the cultivation, the mycelium was separated from the liquid medium through filtration using filters in a Büchner funnel. For the liquid-medium fraction, a liquid–liquid extraction technique was employed using ethyl acetate with a volume ratio of 1:1, and three extractions were performed.

The remaining mycelium portion underwent maceration with twice its volume of ethyl acetate for 24 h. Subsequently, the extracted fractions from both mycelium and broth were combined, and the solvent was evaporated. Thereafter, the extract was preserved at −10 °C for later use in biological tests. Following extraction, the total dry extract obtained from 200 mL of culture media and mycelium was weighed for control purposes.

For chemical profiling, the extracts were subjected to a TLC experiment to identify the main groups of chemical compounds present in the samples. Forty milligrams of the extract were dissolved in 1 mL of ethyl acetate, and 1 µL of the sample, corresponding to 40 µg of extract, was placed on the TLC plate. A mixture of n-hexane and ethyl acetate (1:1) was used as the mobile phase. To detect the chemical classes, ultraviolet light at 254 nm and 365 nm and the following chemical developers were used. To prepare the *p*-anisaldehyde developer, 0.5 mL of *p*-anisaldehyde was mixed with 10 mL of acetic acid, 85 mL of methanol, and 5 mL of concentrated H_2_SO_4_ in that order. The ferric chloride solution was obtained by diluting 3 g of FeCl_3_ in 100 mL of ethyl alcohol. The aluminum chloride solution was prepared by dissolving 1 g of AlCl_3_ in 100 mL of ethyl alcohol. The partition factors (Rf) were calculated as the ratio of the distance migrated by the compound to the distance migrated by the eluent [10,36].

An ultra performance liquid chromatography (UPLC) system (ACQUITY H-Class, Waters, MA, USA) equipped with a quaternary pump, auto-sampler, photodiode array detector, QDa Mass detector, and BEH C18 column (2.1 × 100 mm, 1.7 μm) was utilized for the analysis of major compounds. The gradient elution was performed using solvent A (0.1% formic acid in water) and solvent B (acetonitrile) at a flow rate of 0.4 mL/min as follows: 0–2 min, 2% B; 2–15 min, 2–20% B; 15–20 min, 20–100% B; 20–23 min, 100% B; 23–26 min, 100–2% B; and 26–28 min, 2% B. Detection was set at a wavelength of 230 nm. The column temperature was maintained at 40 °C, the injection volume was 2 µL, and the flow rate was 0.4 mL/min. A QDa mass detector was used together with an electrospray ionization (ESI) source in positive mode under the following conditions: capillary voltage, 1.5 kV; cone voltage, 15 V; sampling frequency, 5 Hz; and probe temperature, 600 °C. The MS full scan was acquired in the range of *m*/*z* 120–1200.

Gas chromatography–mass spectrometry (GC-MS) was performed using a 431-GC and 210-MS system (Varian, Palo Alto, CA, USA) with a FactorFour Capillary column, VF-1 ms (15 m × 0.25 mm). The initial column oven temperature was ramped from 50 °C to 250 °C at a rate of 20 °C/min. Samples diluted with acetonitrile to a concentration of 0.1–0.5 mg/mL were injected in split injection mode at 275 °C, with a split ratio of 10:1.

### 2.5. Antibacterial Test: Determination of Minimum Inhibitory Concentration (MIC), Disk Diffusion, and TLC–Bioautography

For the evaluation of the antimicrobial activity, the minimum inhibitory concentration (MIC) was determined through microdilution tests in extracts according to the Clinical and Laboratory Standard Institute [37,38,39,40], with modifications. All compounds were subjected to MIC tests against three pathogenic species of bacteria. The bacterial strains *Staphylococcus aureus* (KCTC 3881, Gram-positive), *Enterococcus faecalis* (KCTC 2011, Gram-positive), and *Escherichia coli* (KCTC 1039, Gram-negative) were used. Bacterial strains were cultured in each medium. *E. coli* and *E. faecalis* were cultured at 37 °C in aerobic conditions in nutrient medium (MBcell, Seoul, Republic of Korea), and *E. faecalis* was cultured at 37 °C in a fecaltative anaerobic condition in BHI medium (MBcell, Republic of Korea).

The cultured bacterial medium was then diluted until the absorbance at 660 nm reached 0.03 (equivalent to 1–2 × 10^7^ CFU/mL). The microplate wells were inoculated with the bacterial culture medium 20 min before the test.

Fungal extracts were solubilized in each medium at a concentration of 2000 µg/mL, followed by twofold serial dilutions to concentrations of 1000, 500, 250, 125, and 62.5 µg/mL. In a sterilized 96-well plate (Falcon, Dublin, OH, USA), 1 mL of sample was added, and 10 μL of the bacterial culture medium was inoculated. The plate was then incubated for 12 h in a shaking incubator at 150 rpm. Inhibition rates were determined using photometry (BIO-RAD Laboratories Inc., USA) at OD620 nm. Ampicillin was used as the positive control, with a MIC90 of 1.25 µg/mL for *E. coli* and *E. faecalis*, and a range of 20~0.625 µg/mL for *S. aureus* (MIC90 not determined).

The disk diffusion method was used according to the methodology established by the Clinical and Laboratory Standards Institute (CLSI 2015) and the European Committee on Antimicrobial Susceptibility Testing [38]. An *E. coli* culture was spread on Petri dishes containing Mueller–Hinton agar, and filter paper discs treated with extracts (100 µg/disc) were placed on the agar. Antibacterial activity was assessed after 24 to 48 h of incubation at 37 °C. The presence of a clear zone around the discs indicated the antibacterial nature of the endophytic fungal extracts.

For the TLC–bioautography assay, the extracts were applied to TLC plates using the same method as in the chemical profiling experiments. The plates were then overlaid with a layer of *E. coli* on Mueller–Hinton agar, with an initial concentration corresponding to 0.5 on the McFarland scale (1.5 × 10^8^ cells) [41,42]. The inhibition zones on the TLC plates were compared to identify the main groups of chemical compounds, correlating these results with those from the chemical profiling.

## 3. Results

### 3.1. Isolation and Molecular Identification of Endophytic Fungi from Psychotria poeppigiana Müll. Arg. (Rubiaceae)

Five endophytic fungi were successfully isolated from leaves of *P. poeppigiana* collected in the tropical Amazon area, a plant known for its traditional medicinal uses as an analgesic and anti-inflammatory agent. The fungal isolates underwent both morphological and molecular identification processes. All five strains were effectively amplified using primers ITS5/ITS3 for strain 25 and EF4f/Fung5r for strains SMB-23, 24, 26, and 27. BLAST searches revealed their identities as members of three different genera: one *Neopestalotiopsis* sp. (SMB-23), one *Aspergillus* sp. (SMB-27), and three *Penicillium* sp. (SMB-24, SMB-25, and SMB-26), all classified as endophytic fungi (Figure 2, Figure 3, Figure 4, Figure 5 and Figure 6).

Phylogenetic trees constructed using the partial sequence ITS of 18S rRNA gene sequences depicted a close phylogenetic relationship with partial sequences of fungal strains and some fungal species. The challenges in species-level determination and the scarcity of molecular data for comparison underscore the difficulties encountered in identifying many species of endophytic fungi [43]. Figure 2, Figure 3, Figure 4, Figure 5 and Figure 6 present the results of the phylogenetic trees depicting the endophytic fungal strains isolated from *P. poeppigiana* Müll. Arg.

The sequences were analyzed using the neighbor-joining method, and the best-fit model was selected based on the closest sequences for constructing the phylogenetic trees. The sequences obtained in this study were deposited in GenBank (Table 2). Given the low probability and the limited approach to the genus level, it is evident that an alternative locus is required for proper molecular identification and to expand the sequence data (Figure 2, Figure 3, Figure 4, Figure 5 and Figure 6).

### 3.2. Morphological Characteristics of Endophytic Fungi

The description of cultural and morphological characteristics of fungal endophytes, along with microphotographs of their morphological structures, is presented in Table 3 and Figure 7.

Colonies of endophytic fungi typically exhibit rapid growth on PDA medium at 30 °C, ranging between 3 and 7 days. However, growth is comparatively slower on yeast extract sucrose agar (YES) culture medium [28], taking between 7 and 10 days.

The endophytic fungus SMB-23 belongs to the genus *Neopestalotiopsis*, sharing morphological characteristics akin to the genus *Discosia*. When isolated, SMB-23 exhibited a colony with rapid vegetative growth, appearing cottony, irregular, and flat, ranging in color from whitish to gray toward the center, with a lightly colored reverse side and dense mycelial culture (Figure 7, Table 3). However, sporulation was notably slow, which is a result identical with that in the literature [46]. Microscopically, SMB-23 displays septate, hyaline, thin, and dense hyphae, along with cylindrical, fusiform conidia featuring obtuse ends, which are smooth-walled, hyaline, aseptate, and relatively scarce. The presence of appressoria is abundant, with some being globose, others clavate, and some displaying complex structures with irregular lobes; all were aseptate and white in color (Figure 7, Table 3). Generally, the color of the median conidial cells facilitates the differentiation of three genera: *Neopestalotiopsis*, *Pestalotiopsis*, and *Pseudopestalotiopsis* [46]. Conidia with versicolored median cells are characteristic of the genus *Neopestalotiopsis*, which is believed to have evolved from the lineage of *Pseudopestalotiopsis*, characterized by dark concolorous conidial median cells, while *Pestalotiopsis* typically presents three light concolorous conidial median cells [46]. The morphology of *Pestalotiopsis*-like taxa exhibits variability depending on the environment and host from which they were isolated, rendering species separation based on phenotypic characteristics challenging. *Neopestalotiopsis* can be easily distinguished from *Pestalotiopsis* by its fusiform conidia with five cells and versicolored median cells [47].

The three strains belonging to the *Penicillium* genus exhibit rapid colony growth, initially appearing white and later developing a yellowish-green hue with a creamy yellow reverse side. Their texture is flat, filamentous, and either velvety or cottony. SMB-24 and SMB-26 display drops of exudate after 10 days of growth on PDA. The colonies of *Penicillium* strains SMB-24, 25, and 26 possess a cotton-like texture, albeit with varying shades; SMB-24 and 26 tend towards a more yellow-green hue, while SMB-25 appears greenish (Figure 7, Table 3). In contrast, strain SMB-23 exhibits a rough surface texture, with aerial mycelium resembling cotton in white layers.

Strain *Aspergillus* SMB-27 displays abundant black conidia on the surface and lacks a cottony appearance (Figure 7). For the genus *Aspergillus*, ascospore sizes and morphology—especially diagnostic ornamentation such as roughening, rims, wings, and furrows—play a crucial role in species identification [30]. Strain SMB-27 of *Aspergillus* forms filamentous hyphae, resembling miniature plants. The initial mycelium color is white and contains growth which turned black within two days on PDA, accompanied by the production of conidial spores. The colony’s reverse side appears yellow.

### 3.3. Chemical Profile of Endophytic Fungi Extracts

The secondary metabolites were extracted from five endophytic fungi using ethyl acetate. Cultures of *Neopestalotiopsis* sp. SMB-23, *Penicillium* sp. SMB-24, sp. SMB-25, and sp. SMB-26 and *Aspergillus* sp. SMB-27 were grown for 15 days in 200 mL of medium, resulting in extract yields of 31.8 mg, 138.6 mg, 201.9 mg, 161.9 mg, and 12.2 mg, respectively. In cultures of 9, 12, 15, 18, and 30 days for the endophytic fungi, similar metabolite profiles in TLC were obtained after 12 days. The chemical profiles of the extracts were investigated using thin-layer chromatography (TLC) and staining methods to identify the main groups of chemical compounds (Figure 8).

The TLC analysis did not reveal any chemical compounds with chromophores or high unsaturation visible under normal light (Figure 8A). However, the presence of conjugated double bonds was observed under UV light at 254 nm in samples from SMB-24, SMB-25, and SMB-26, with an Rf value of 0.35 (Figure 8B). Flavonoid compounds, stained with aluminum chloride, were observed in all extracts under UV light at 365 nm (Figure 8C). After developing the TLC plate with p-anisaldehyde, purple spots were observed in all extracts at Rf values of 0.96 and 0.88, indicating the presence of terpenes (Figure 8D). Additionally, red spots at Rf 0.60 in SMB-25 and SMB-26, and at Rf 0.35 in all three *Penicillium* extracts (SMB-24, SMB-25, and SMB-26), indicated the presence of flavonoids. These flavonoids were also observed under UV light at 254 nm (Figure 8B), showing conjugated double bonds and corroborating the identified compounds. The presence of phenolic compounds was indicated by brown spots developed with ferric chloride in extracts SMB-23, SMB-24, SMB-25, and SMB-26, all at an Rf value of 0.58 (Figure 8E).

To confirm the presence of phenolic compounds, a UPLC analysis was conducted at 230 nm, which is known for its high absorbance for phenolic compounds (Figure 9). Representative results from strains SMB-25 and SMB-26 displayed similar UPLC profiles (Figure 9A,B) with the highest UV absorbance at 236 nm and 227 nm, respectively (Figure 9C,D). The molecular ion [M+H]^+^ of the two peaks were measured as 170.95 and 154.93 *m*/*z*, respectively (Figure 9E,F). These measurements suggest the presence of tri-hydroxy benzoic acid (C_7_H_6_O_5_, 172.12 g/mol calculated, e.g., gallic acid and phloroglucinol carboxylic acid) and di-hydroxy benzoic acid (C_7_H_6_O_4_, 154.12 g/mol calculated, e.g., orsellinic acid and gentisic acid) as metabolites from *Penicillium* genera [48,49].

Volatile metabolites were further investigated via GC-MS (Figure 10). As metabolites of mono-terpen and phenolic and fatty acids from *Penicillium* genera [50,51,52,53] were detected, these results suggest the presence of sulcatone (C_8_H_14_O, 126.20 g/mol calculated), orcinol (C_7_H_8_O_2_, 124.14 g/mol calculated), (2Z or 2E)-2-butendioic acid 2-(1-methyl ester) (C_8_H_10_O_4_, 170.16 g/mol calculated), 1-hexadecanoic acid (C_16_H_32_O_2_, 256.42 g/mol calculated), and 9,12-octadecadienoic acid (C_18_H_32_O_2_, 280.44 g/mol calculated) (Figure 10B–F).

### 3.4. Antimicrobial Activity of Extracts from Endophytic Fungi

The extracts were assessed for antimicrobial activity against pathogenic bacteria, including *Staphylococcus aureus* (Gram-positive), *Escherichia coli* (Gram-negative), and *Enterococcus faecalis* (Gram-positive), by determining their minimum inhibitory concentrations (MICs, mg/mL).

The extracts from *Penicillium* SMB-24, SMB-25, and SMB-26 exhibited potent antibacterial activity against all three pathogenic bacteria, with MIC values ranging from 0.0625 to 2.0 mg/mL. Conversely, the extracts from *Neopestalotiopsis* SMB-23 and *Aspergillus* SMB-27 showed no activity up to 2.0 mg/mL (Table 4 and Table 5). Notably, *Penicillium* SMB-24, SMB-25, and SMB-26 demonstrated higher efficacy against *Staphylococcus aureus* and *Escherichia coli* (MIC: 0.0625–0.5 mg/mL) compared to *Enterococcus faecalis* (MIC: 1.0–2.0 mg/mL). These findings suggest that *Penicillium* SMB-24, SMB-25, and SMB-26 possess antibacterial properties against both Gram-positive and Gram-negative pathogenic bacteria (Table 4 and Table 5).

### 3.5. Relationship between Antibacterial Activity and Chemical Profile

Since *Penicillium* SMB-24, SMB-25, and SMB-26 possess antibacterial properties against both Gram-positive and Gram-negative pathogenic bacteria (Table 4 and Table 5), the extract from these strains were subjected to identify the relationship between antibacterial activity and chemical profile classified (Figure 11). As a control, the antimicrobial activity was confirmed using Gram-positive *E. coli*, and showed clear zones in all three strains and positive control (Figure 11A). In the TLC–bioautography assay, the area of terpene (around Rf 0.35) showed high activity against *E. coli* for the extracts of endophytic fungi, *Penicillium* species (SMB-24, 25 and 26) (Figure 11B,C). Therefore, this observation and relationship can propose that the responsible active compound is classified as a terpene compound. Furthermore, *Penicillium* species can be a rich source of the terpene compound, which was present in high amounts in TLC (Figure 8D).

Since *Penicillium* SMB-24, SMB-25, and SMB-26 exhibited antibacterial properties against both Gram-positive and Gram-negative pathogenic bacteria (Table 4 and Table 5); their extracts were analyzed to identify the relationship between antibacterial activity and chemical profile (Figure 11).

As a control, the antimicrobial activity was confirmed using Gram-negative *E. coli*, which showed clear inhibition zones for all three strains and the positive control (Figure 11A). In the TLC–bioautography assay, the terpene region (around Rf 3.5) demonstrated high activity against *E. coli* for the extracts from the *Penicillium* species SMB-24, SMB-25, and SMB-26 (Figure 11B,C). This observation suggests that the active compound responsible for the antibacterial activity is likely a terpene. Furthermore, the *Penicillium* species appear to be rich sources of terpene compounds, as indicated by the high content observed in the TLC analysis (Figure 8D). Meanwhile, the extracts from both the mycelium and the medium showed similar chemical profiles in TLC analysis, especially in the extracts of *Penicillium* species; the mycelium contained 1–2 times more the amount of the extracts.

## 4. Discussion

A species of Rubiaceae, *P. poeppigiana* Müll. Arg. (accepted as *Palicourea tomentosa* (Aubl.) Borhidi), was collected in the Amazon region of Bolivia. Bolivia is renowned for its rich biological diversity, boasting a vast array of plant species estimated to range between 17,000 and 20,000 vascular plants, thus contributing significantly to global biodiversity.

The indigenous communities of Bolivia possess profound knowledge of the medicinal properties of the surrounding flora, passed down through generations via oral traditions and direct observation. *P. poeppigiana* Müll. Arg. has long been utilized in traditional medicine for treating inflammation and pain, with *Cephaelis elata* serving as a synonym for dementia treatment [21].

Recent studies have unveiled the antioxidant, anti-inflammatory, and anti-acetylcholinesterase (AChE) activities of essential oil derived from *P. poeppigiana* leaves (EOPP) [21]. However, the endophytic fungi associated with *P. poeppigiana* have not been thoroughly explored, particularly concerning their antibacterial properties. The traditional use of the plant and its ecological habitat are crucial factors to consider when isolating endophytes, as regions with high biodiversity harbor endophytes with potentially diverse properties [23,24].

In this study, five novel species of endophytic fungi were isolated from the leaves of *P. poeppigiana* Müll. Arg. and identified as *Neopestalotiopsis* sp. SMB-23, *Aspergillus* sp. SMB-27, and *Penicillium* sp. SMB-24, SMB-25, and SMB-26. Furthermore, the extracts obtained from these fungi cultured in potato dextrose broth (PDB) were evaluated for antibacterial activity, specifically the minimum inhibitory concentration (MIC), against *Escherichia coli*, *Staphylococcus aureus*, and *Enterococcus faecalis*. Among them, the extracts from the three *Penicillium* species exhibited potent antibacterial activity against all three pathogens (both Gram-positive and Gram-negative bacteria), with MIC values ranging from 62.5 to 2000 µg/mL. Additionally, the study on the relationship between antibacterial activity and TLC-bioautography suggested that the active compound responsible for the observed antibacterial activity is a terpene. There is also a possibility of a synergistic effect of the metabolites, as non-selective solvent extracts often exhibit such interactions.

Based on the GC-MS analysis, the microbial active metabolite responsible for the observed activity was proposed to be a monomeric terpene, (2Z)-2-butenedioic acid, 2-(1-methylethenyl)-, and 4-methyl ester. This compound was also found in the extract from *Aspergillus sclerotiorum* AS-75 [50] and has demonstrated inhibitory effects on rice bacterial blight, similarly observed in other *Penicillium* species [51]. Also, a biosynthetic compound, sulcatone (6-methyl-5-hepten-2-one) from geraniol from *Penicillium digitatum,* showed to be an effective insecticide against *Spodoptera littoralis*, with a strong impact on larval mortality [53,54].

The *Penicillium* species SMB-24, SMB-25, and SMB-26 demonstrated high terpene content in the TLC analysis (Figure 8D). These species produced significant yields of the active compound, with extract yields of 138.6 mg, 201.9 mg, and 161.9 mg, respectively, indicating their potential as rich sources of terpenes on a large scale. Furthermore, the secretion of an active compound, a terpen, in the medium suggests the potential for developing a continuous process in a bioreactor, where metabolites can be harvested directly from the medium.

Over the past three decades, the rise in antimicrobial resistance among pathogenic microbes has posed significant challenges, compelling scientists, professionals, and clinical specialists to seek new solutions [7,15]. Certain bioactive metabolites with unique structures have been identified within endophytic fungi, holding promise for combating increasing antimicrobial resistance [6]. For instance, ethyl acetate extracts of *F. lateritium* and *Xylaria* sp., isolated from *Rhizophora mucronata*, demonstrated inhibition against the tested bacterial pathogens at varying concentrations [7]. For these two fungi, crude extracts at 500 µg/mL were sufficient for inhibiting *E. coli* and *P. aeruginosa*, while 1000 µg/mL was needed to inhibit *S. aureus*, and the highest concentration, at 2000 µg/mL, was required for *B. subtilis*.

Previous reports have also demonstrated similar results with *Penicillium* species. For instance, ethyl acetate extracts of endophytic *Penicillium lanosum* (PL) and *Penicillium radiatolobatum* (PR) showed minimum inhibitory concentrations (MICs) ranging from 31.25 to 500 µg/mL against Gram-positive and Gram-negative pathogens such as *B. cereus*, *S. aureus*, *L. monocytogenes*, *E. coli*, and *S. enterica* [55]. Additionally, Mosquera (2020) reported the antibacterial activity of an endophytic fungus of the genus *Penicillium* spp. isolated from *M. americana* and *M. oleifera* against *E. coli* and *S. aureus*, with MICs of 1000 and 2000 µg/mL, respectively [56]. Another strain of endophytic fungi, *Penicillium* sp. isolated from the mangrove plant (*Rhyzopora mucronata*), exhibited antibacterial activity against *S. aureus* ATCC 9144 and *E. coli* ATCC 8739 [57].

However, extracts from *Neopestalotiopsis* SMB-23 and *Aspergillus* SMB-27 showed no activity up to the concentration of 2000 µg/mL in this study. *Neopestalotiopsis* was initially grouped with *Pestalotiopsis* but was re-classified as a separate genus based on morphological and DNA data in 2014 [46,58]. A secondary metabolite, Neopestalotin B, purified from the endophytic fungus *Neopestalotiopsis* sp., exhibited inhibitory effects against Gram-positive bacteria, including *B. subtilis*, *S. aureus*, and *S. pneumoniae*, with MIC values of 10, 20, and 20 μg/mL, respectively [58]. Additionally, chloroform–methanol (1:1) crude extracts of *Neopestalotiopsis* and *Pestalotiopsis* species isolated from the *Manilkara zapota* plant were found to be inactive against *E. coli* (ATCC 25922 and ATCC 35218), *S. aureus*, and *E. faecalis* [59].

Meanwhile, an endophytic fungus, *Aspergillus* sp. EJC08, isolated from the medicinal plant *Bauhinia guianensis*, exhibited higher activity against *B. subtilis* with its hexane extract, while the methanol extract showed greater efficacy against *S. aureus*. The isolated pure-alkaloid substances, including fumigaclavine C and pseurotin A, were approximately 10 times more potent than the ethyl acetate extract from which they were derived, with MIC values of 7810 µg/mL [59]. Consequently, the antimicrobial activity of crude extracts from *Neopestalotiopsis* and *Aspergillus* species may generally be lower compared to *Penicillium* species.

Filamentous fungi of the *Aspergillus* genus are characterized by their filament-like cell chains, known as hyphae. Given their ability to release multiple enzymes and their role in fermentations, the peptides they produce are of considerable interest and warrant exploration and study. The biotechnological potential of the *Aspergillus* genus is therefore underscored, as these microorganisms play a crucial role in producing molecules and enzymes of scientific and pharmaceutical significance, as well as in various industries, particularly in food production [30].

These findings highlight the potential of these endophytic fungi, particularly *Penicillium* species, as valuable sources of secondary metabolites with significant biological activities. This suggests promising applications in medicine, pharmaceuticals, agriculture, and environmental technologies.

## 5. Conclusions

In conclusion, this study successfully isolated and identified five novel endophytic fungi from the leaves of *P. poeppigiana* Müll. Arg, a plant belonging to the Rubiaceae family. These fungi were identified as *Neopestalotiopsis* sp. SMB-23, three *Penicillium* sp. SMB-24, 25, and 26, and *Aspergillus* sp. SMB-27. The extracts obtained from these fungi, cultured in PDB, were evaluated for their antibacterial activity (MIC) against *E. coli*, *S. aureus*, and *E. faecalis*. Remarkably, extracts from the three *Penicillium* species exhibited potent antibacterial activity against all three pathogens, including both Gram-positive and Gram-negative bacteria, with MIC values ranging from 62.5 to 2000 µg/mL. Conversely, extracts from *Neopestalotiopsis* and *Aspergillus* species showed no activity up to a concentration of 2000 µg/mL.

This study has laid the groundwork for further exploration by enabling the isolation of bioactive endophytic fungi from the *P. poeppigiana* Müll. Arg. plant. It opens avenues for future research to delve into the chemical isolation and elucidation of the secondary metabolites responsible for their activity. Further studies are needed to determine optimal production conditions for active compounds, as factors such as time, temperature, light–dark exposure, and medium composition can significantly influence metabolite production. This can be applied to both batch and continuous processes in a bioreactor, where metabolites can be harvested directly from media. Additionally, deeper identification at the species level could shed light on the importance of fungus–plant associations in the quest for new active compounds. Further tests will be conducted to check for mycotoxins and other bioactivities such as antioxidants, cytotoxicity, and anti-diabetic effects.

Therefore, these findings underscore the potential of these endophytic fungi, particularly the *Penicillium* species, as valuable sources of secondary metabolites with significant biological activities. This suggests promising applications in various fields, including medicine, pharmaceuticals, agriculture, and environmental technologies.

## Figures and Tables

**Figure 1 microorganisms-12-01590-f001:**
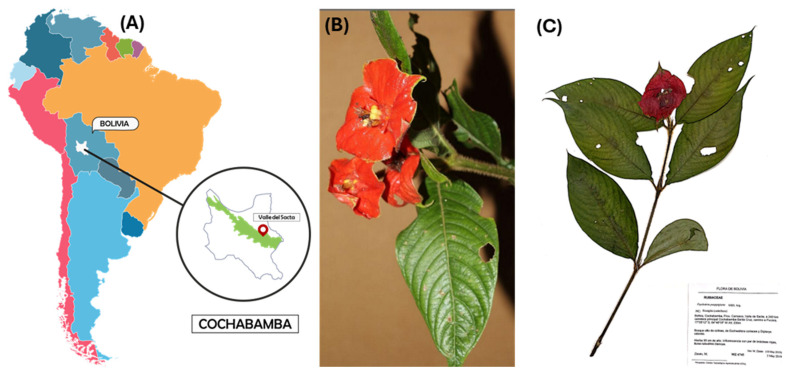
Collection of plant species. (**A**) Location in the Amazon region called Valle del Sacta at 240 m above sea level (17°05′12″ S and 64°46′19″ W), Bolivia. (**B**) *P. poeppigiana* Müll. Arg. (**C**) Micro-herbarium preparation, specimen assembly.

**Figure 2 microorganisms-12-01590-f002:**
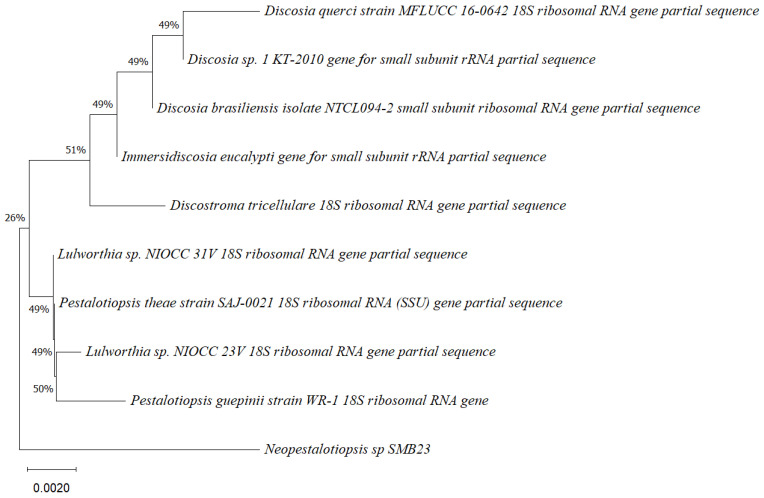
Phylogenetic tree based on the partial sequence of a small-subunit ribosomal RNA gene of the endophytic fungus *Neopestalotiopsis* sp. SMB-23 (accession no. PP800334) obtained with EF4f/Fung5r, showing its relationship via neighbor-joining with other closely related taxa from NCBI GenBank. The scale bar indicates nucleotide substitutions per site, using the neighbor-joining method. The numbers of the nodes indicate the bootstrap values of 1000 replicates. The model used was Kimura 2 (K2). The tree was rooted in *Neopestalotiopsis* clavispora18S ribosomal RNA gene. SMB-23 was identified as *Neopestalotiopsis* sp. based on its morphological traits.

**Figure 3 microorganisms-12-01590-f003:**
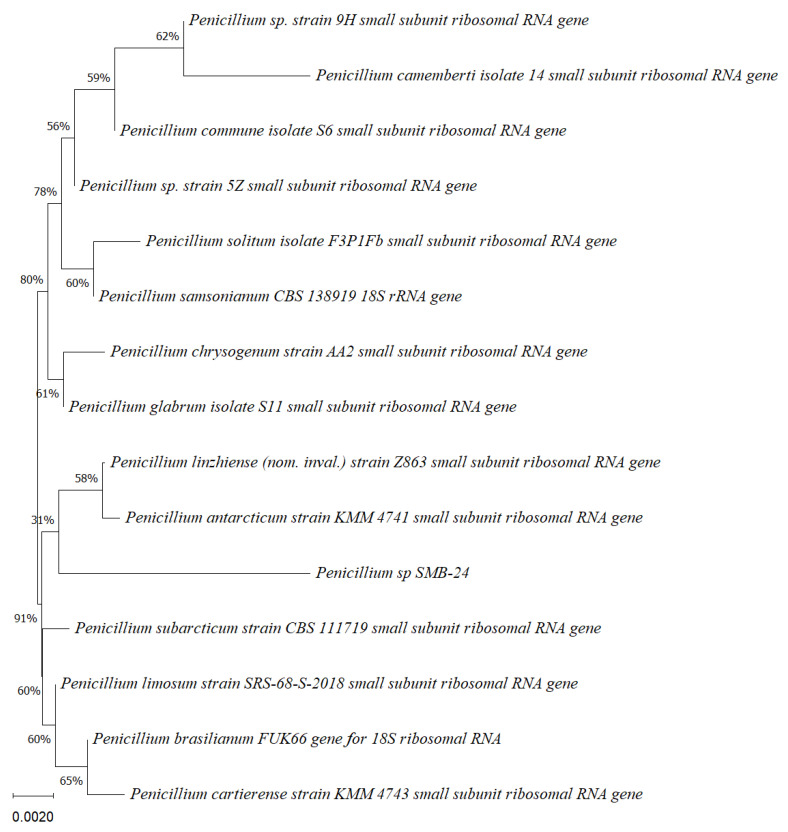
Phylogenetic tree based on the partial sequence of a small-subunit ribosomal RNA gene of the endophytic fungus *Penicillium* sp. SMB-24 (Accession no. PP800436) obtained with EF4f/Fung5r primers, showing a relationship via neighbor-joining with other closely related taxa from the NCBI GenBank. The scale bar indicates nucleotide substitutions per site, using the neighbor-joining method. The numbers of the nodes indicate the bootstrap values of 1000 replicates. The model used was Jukes–Cantor (JC).

**Figure 4 microorganisms-12-01590-f004:**
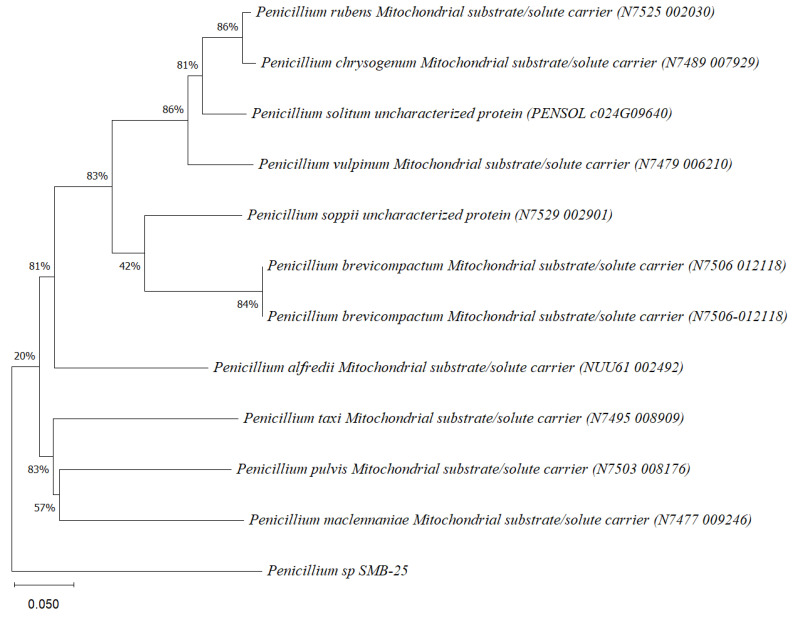
Phylogenetic tree based on the sequence of mitochondrial substrate/solute carrier gene of the endophytic fungus *Penicillium* sp. SMB-25 obtained with ITS5f/ITS3r primers, showing a relationship by neighbor-joining with other closely related taxa from NCBI GenBank. The scale bar indicates nucleotide substitutions per site, using the neighbor-joining method. The numbers of the nodes indicate the bootstrap values of 1000 replicates. The model used was Tajima–Nei (TN).

**Figure 5 microorganisms-12-01590-f005:**
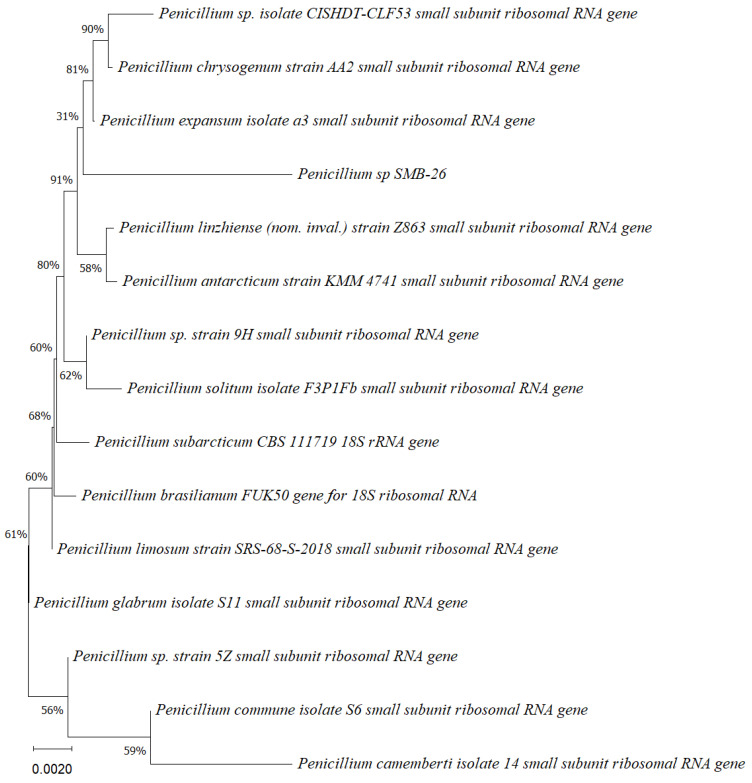
Phylogenetic tree based on the partial sequence of small-subunit ribosomal RNA gene of the endophytic fungus *Penicillium* sp. SMB-26 (accession no. PP800338) obtained with EF4f/Fung5r primers, showing a relationship via neighbor-joining with other closely related taxa from NCBI GenBank. The scale bar indicates nucleotide substitutions per site, using the neighbor-joining method. The numbers of the nodes indicate the bootstrap values of 1000 replicates. The model used was Jukes–Cantor (JC).

**Figure 6 microorganisms-12-01590-f006:**
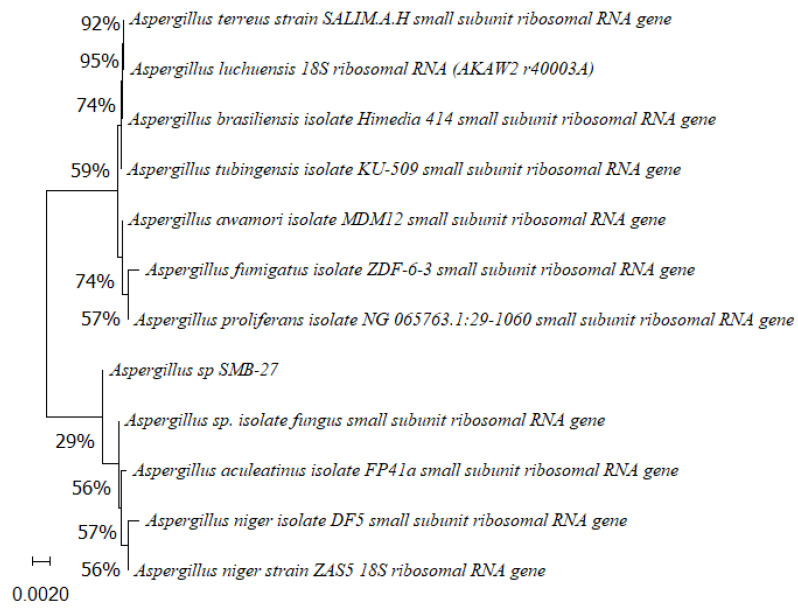
Phylogenetic tree based on the partial sequence of small-subunit ribosomal RNA gene of the endophytic fungus *Aspergillus* sp. SMB-27 (accession no. PP800435) obtained with EF4f/Fung5r primers, showing a relationship via neighbor-joining with other closely related taxa from NCBI GenBank. The scale bar indicates nucleotide substitutions per site, using the neighbor-joining method. The numbers of the nodes indicate the bootstrap values of 1000 replicates. The model used was Jukes–Cantor (JC).

**Figure 7 microorganisms-12-01590-f007:**
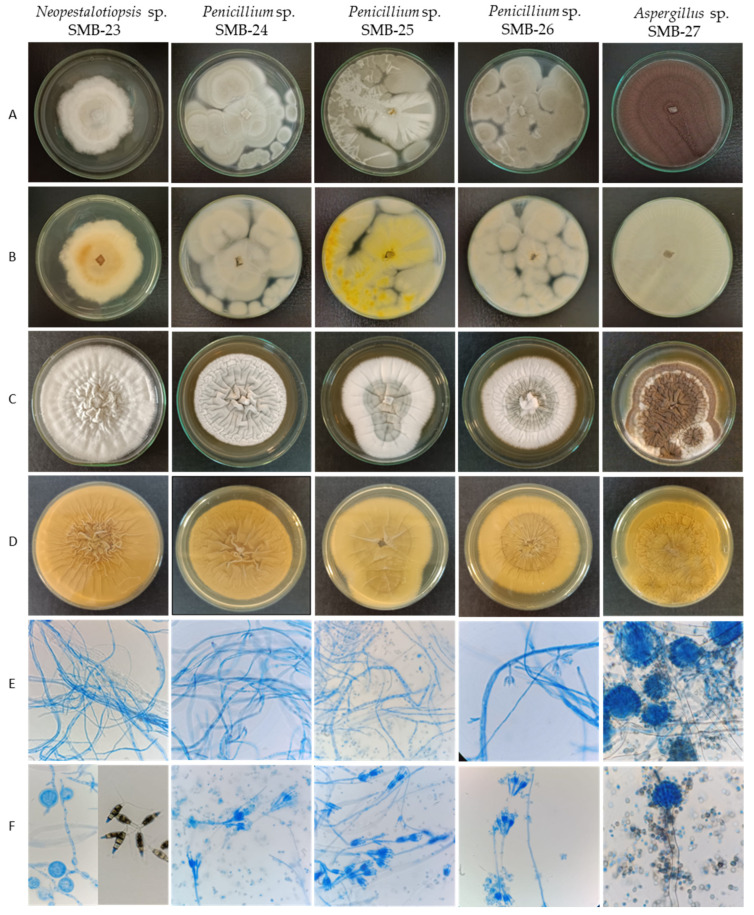
Macro- and micromorphological characteristics of endophytic fungi from *Psychotria poeppigiana* Müll. Arg. (LRI-OLYMPUS-100×/0.65): Colonies on PDA’s upper side (**A**) and reverse side (**B**) after 7 days at 30 °C. Colonies in YES upper side (**C**) and reverse side (**D**) after 7 days at 30 °C. Generative hyphae and mycelium (**E**) and conidiophores and conidia (**F**).

**Figure 8 microorganisms-12-01590-f008:**
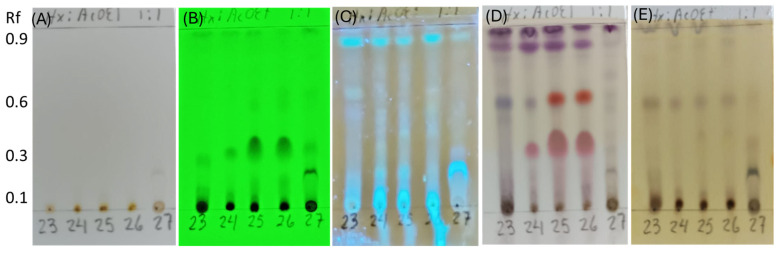
Thin-layer chromatography (TLC) of extracts from endophytic fungi strains, SMB-23, SMB-24, SMB-25, SMB-26, and SMB-27 (23, 24, 25, 26, and 27). (**A**) Compounds with chromophores or high unsaturation observed under visible light. (**B**) Presence of conjugated double bonds observed under UV light at 254 nm. (**C**) Presence of flavonoids stained with aluminum chloride and illuminated under UV light at 365 nm. (**D**) Compounds stained with p-anisaldehyde, indicating the presence of terpenes (purple spots) and flavonoids (red spots). (**E**) Phenolic compounds stained with ferric chloride.

**Figure 9 microorganisms-12-01590-f009:**
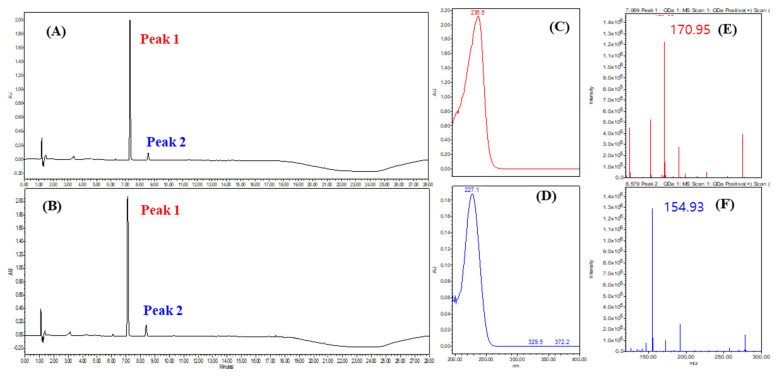
Representative UPLC chromatogram of (**A**) SMB-25 and (**B**) SMB-26 at 230 nm. (**C**) UV spectrum of peak 1. (**D**) UV spectrum of peak 2. QDa positive scan TIC of (**E**) peak 1 and (**F**) peak 2 analyzed via single mass spectrometry.

**Figure 10 microorganisms-12-01590-f010:**
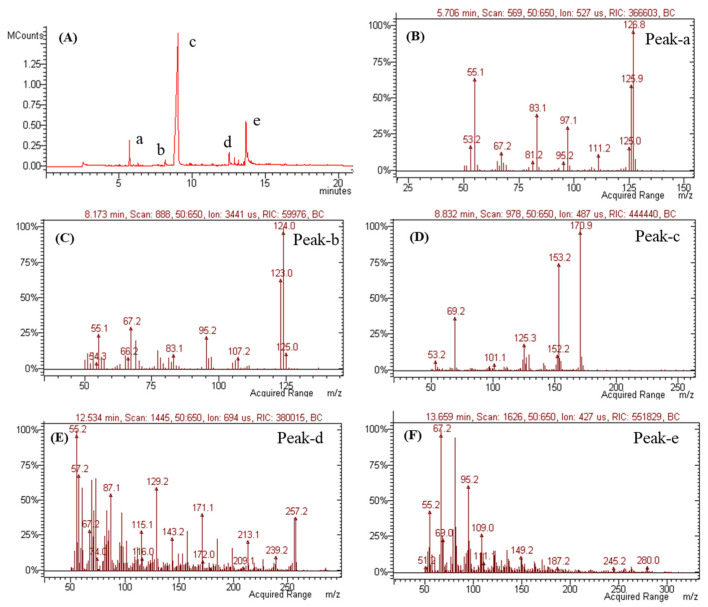
GC-MS analysis of SMB-25. (**A**) GC chromatogram and mass spectra of (**B**) peak-a, (**C**) peak-b, (**D**) peak-c, (**E**) peak-d, and (**F**) peak-e in chromatograms.

**Figure 11 microorganisms-12-01590-f011:**
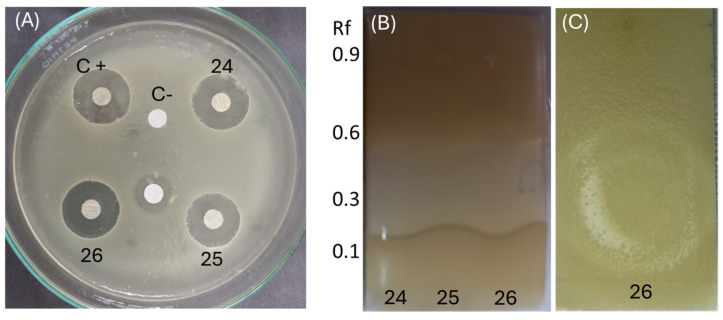
Relationship between antibacterial activity and chemical profile. (**A**) Disc diffusion assay using *E. coli* to evaluate the antibacterial activity of extracts from *Penicillium* species SMB-24 (24), SMB-25 (25), and SMB-26 (26). (**B**,**C**) TLC-bioautography assay demonstrating the antibacterial activity of these extracts against *E. coli*, highlighting the active regions, particularly around Rf 0.35.

**Table 1 microorganisms-12-01590-t001:** Molecular markers and the PCR primers and programs used.

Loci	PCR Primers	Sequence (5′–3′)	PCR Cycles			Ref.
			Denaturation	Annealing	Polymerization	
ITS	F: ITS5 R: ITS3	GGA AGT AAA AGT CGT AAC AAG G GCA TCG ATG AAG AAC GCA GC	94 °C: 2 min, 35 cycles 94 °C: 30 s	53 °C: 30 s	72 °C: 1 min 72 °C: 1 min 4 °C: ∞	[33]
TEF1-α	F: EF4f R: Fung5r	GGA AGG GGA TGT ATT TAT TAG GTA AAA GTC CTG GTT CCC	94 °C: 2 min, 35 cycles 94 °C: 30 s	53 °C: 30 s	72 °C: 1 min 72 °C: 1 min 4 °C: ∞	[32]

F = forward primer; R = reverse primer.

**Table 2 microorganisms-12-01590-t002:** GenBank accession numbers for the endophytic fungi isolates from *P. poeppigiana* Müll. Arg.

Species	Source	GenBank Accession Number, ITS
*Neopestalotiopsis* sp. SMB-23	Leaf	PP800334
*Penicillium* sp. SMB-24	Leaf	PP800436
*Penicillium* sp. SMB-25	Leaf	2829790 ^a^
*Penicillium* sp. SMB-26	Leaf	PP800338
*Aspergillus* sp. SMB-27	Leaf	PP800435

^a^ Submission I.D.

**Table 3 microorganisms-12-01590-t003:** Cultural characteristics of 5 endophytic fungi strains from *Psychotria poeppigiana* Müll. Arg.

Macroscopic and Microscopic Characteristics						
Endophytic Fungi Strains	Colonies (PDA)	Colonies (YES)	Size Colonies	Hyphae	Conidiophores and Conidia	References
*Neopestalotiopsis phangngaensis*	Fluffy to cottony, irregular shape, dull surface, undulate edge, fluffy margin. Upper view white and the reverse primrose.		Colonies reached 75–90 mm after seven days on PDA at 28 °C	Filamentous and septate hyphae with a diameter between 4 and 6 μm, ending in rounded conidiophores.	Conidiogenous cells subcylindrical to ampulliform, hyaline, entroblastic, and thin-walled. Conidia fusiform, straight or slightly curved, basal cell conical, hyaline to pale brown, 3–6 µm long.	[31,44,45]
*Neopestalotiopsis* sp., SMB-23	White concentric growth cottony texture. Rough yellowish.	White layered concentric growth cottony texture. Rough yellowish on the reverse.	Colonies reached 90 mm after the fifth day in PDA at 30 °C and 90 mm in seven days in YES at 30 °C.	Insulated branched. Filamentous and septate hyphae ending in rounded conidiophores.	Oval-shaped conidia with transverse segments with brown coloration with antenna-shaped tips.	This study
*Aspergillus niger*	The growth is initially white but changes to black after a few days, producing conidial spores. The edges of the colonies appear pale yellow, producing radial fissures.		Cottony appearance, initially white to yellow and then turning black. The reverse rough yellow on PDA. In YES, the color is and more orange on the reverse.	Filamentous fungus that forms filamentous hyphae that make them look like small plants.	The conidial heads appear radial, they are smooth and hyaline. The conidiophore becomes dark at the apex and terminating in a globose vesicle which is 30–75 μm in diameter. Produce conidia of brown coloration, and have a diameter of 4–5 μm.	[30,44]
*Aspergillus* sp., SMB-27	Cottony appearance, initially white to yellow and then turning black. The reverse is rough yellowish.	Brown with white concentric edges. Orange rough appearance on the reverse.	Colonies reached 70 mm after the fifth day in PDA at 30 °C and 90 mm in five days in YES at 30 °C.	Insulated, branched.	Conidiophores, globose vesicle, small round conidia.	This study
*Penicillium linzhiense*	White colonies that turn greenish with yellowish parts. Cottony texture.	The colonies are white with a slight greenish coloration and a cottony texture.	Colonies in MEA at 25 °C. After 7 days, the growth varies between 30 and 50 mm in diameter. In PDA, the diameter is larger, up to 60 mm,	It has septate hyaline hyphae.	Conidiophore 20–100 × 2–2.5 μm, occurred in aerial or dragging hyphae with smooth walls. Conidia spherical or subspherical in shape, 2.6–4.5 μm. Conidial chains loose, nearly cylindrical, or irregular.	[28,44]
*Penicillium* sp. SMB-24	Greenish-yellow with white dots and streaks. Cottony, yellowish, striated.	Green, cottony whitish. Rough yellowish.	Colonies reached 70 mm after the fifth day in PDA at 30 °C and 70 mm in seven days in YES at 30 °C.	It has septate hyaline hyphae.	Short conidiophores, in the shape of small trees, with branches called phialides in numbers of 3 or 4. Cylindrical conidia.	This study
*Penicillium* sp. SMB-25	Greenish, cottony texture. Greenish-yellow, flat.	Yellowish-green, cottony whitish. Greenish-yellow light edges.	Colonies reached 80 mm after the fifth day in PDA at 30 °C and 80 mm in seven days in YES at 30 °C.	It has septate hyaline hyphae.	Short conidiophores, in the shape of small trees, with branches called phialides in numbers of 3 or 4. Cylindrical conidia. Conidiophores smaller than in 24 and 26.	This study
*Penicillium* sp. SMB-26	Greenish-yellow with white dots. Cottony, yellowish, flat	Green, cottony whitish. Rough yellowish.	Colonies reached 70 mm after the fifth day in PDA at 30 °C and 90 mm in seven days in YES at 30 °C.	It has septate hyaline hyphae.	Short conidiophores, in the shape of small trees, with branches called phialides in numbers of 3 or 4. Cylindrical conidia.	This study

**Table 4 microorganisms-12-01590-t004:** Minimum inhibitory concentration (MIC) and inhibition activity at various concentrations of crude extract prepared from endophytic fungi against 3 pathogen bacteria.

Endophytic Fungi	Pathogenic Bacteria	Inhibition (µg/mL)					
		2000	1000	500	250	125	62.5
*Neopestalotiopsis* sp. SMB-23	*S. aureus*	-	-	-	-	-	-
	*E. coli*	-	-	-	-	-	-
	*E. faecalis*	-	-	-	-	-	-
*Penicillium* sp. SMB-24	*S. aureus*	+	+	+	-	-	-
	*E. coli*	++	+	+	+	-	-
	*E. faecalis*	++	+	-	-	-	-
*Penicillium* sp. SMB-25	*S. aureus*	++	++	+	-	-	-
	*E. coli*	+++	++	++	++	++	+
	*E. faecalis*	++	-	-	-	-	-
*Penicillium* sp. SMB-26	*S. aureus*	++	++	++	+	+	-
	*E. coli*	++	++	++	++	++	+
	*E. faecalis*	++	+	-	-	-	-
*Aspergillus* sp. SMB-27	*S. aureus*	-	-	-	-	-	-
	*E. coli*	-	-	-	-	-	-
	*E. faecalis*	-	-	-	-	-	-

+++: indicates the sample completely inhibited bacterial growth. ++: indicates the sample completely inhibited bacterial growth by more than 90%. +: indicates the sample completely inhibited bacterial growth by more than 70%. -: indicates no or less than 70% inhibition.

**Table 5 microorganisms-12-01590-t005:** Summary of minimum inhibitory concentration (MIC) and inhibition activity at 250 µg/mL of extract prepared from endophytic fungi.

Endophytic Fungi	MIC (µg/mL)			Inhibition (%) at 250 µg/mL		
	*S. sureus*	*E. coli*	*E. faecalis*	*S. sureus*	*E. coli*	*E. faecalis*
*Neopestalotiopsis* sp. SMB-23	-	-	-	-	-	-
*Penicillium* sp. SMB-24	500	250	1000	62.3	87.1	61.8
*Penicillium* sp. SMB-25	500	62.5	2000	-	97.7	61.0
*Penicillium* sp. SMB-26	125	62.5	1000	83.1	96.4	59.4
*Aspergillus* sp. SMB-27	-	-	-	-	-	-

## Data Availability

The original contributions presented in the study are included in the article, further inquiries can be directed to the corresponding authors.

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
