# Peer review of "Identification, Characterization, and Antibacterial Evaluation of Five Endophytic Fungi from Psychotria poeppigiana Müll. Arg., an Amazon Plant"

_microorganisms, 2024, doi:10.3390/microorganisms12081590_

Round 1

Reviewer 1 Report

Comments and Suggestions for Authors

This publication seems to be within the scope of journal. However, it needs several corrections to be more acceptable for publication.

Chemical analysis of the composition of extracts is absolutely insufficient by modern standards. TLC analyzes are only valid when performed by an automated HP-TLC system (automatic applicator, automatic developing chamber, automatic reader), because then they provide truly reproducible results. When TLC analyzes are performed manually, they do not provide quantitative results, but only a very indicative insight into the composition of the mixture. Currently, the standard for determining the chemical composition of extracts is the use of the HPLC-MS technique. Spectrophotometric determinations are often performed as preliminary tests, e.g. determination of phenolic compounds using the Folin-Ciocalteu method, of course using appropriate standards. In turn, terpenoids are identified using GC-MS with the Kovats index, similarly to fats. It is also important to check whether isolated strains (especially Aspergillus sp.) do not produce mycotoxins.

The text of the publication lacks information why the breeding process was carried out for 15 days and no longer or shorter. It would be advisable to include a growth curve in the publication.

If the mycelium and the medium were extracted separately, I do not understand why these extracts were then combined together. A separate chemical analysis of the medium could reveal metabolites secreted into it. If such compounds were observed, it would significantly increase the value of the publication, because it would create opportunities to develop a continuous process in a bioreactor.

Line 596 It should be „Gram-positive and Gram-negative” instead of „gram-positive and gram-negative”, because it came from the name of the method inventor: Hans Christian Gram, who developed the technique in 1884.

Author Response

Response to reviewers

Manuscript ID: microorganisms-3100339

Title: "Identification, Characterization and Antibacterial Evaluation of Five Endophytic Fungi from Psychotria poeppigiana Müll. Arg. an Amazon Plant"

Reviewer 1:

Comment: This publication seems to be within the scope of journal. However, it needs several corrections to be more acceptable for publication.

Response: Thank you very much for your supportive review comments on our manuscript. We appreciate your valuable feedback and have made the necessary corrections to improve the manuscript. Please find the detailed responses below and the corresponding revisions highlighted in the re-submitted manuscript. We agree with all your comments, which are valuable for our future work on this project.

Comment: Chemical analysis of the composition of extracts is absolutely insufficient by modern standards. TLC analyzes are only valid when performed by an automated HP-TLC system (automatic applicator, automatic developing chamber, automatic reader), because then they provide truly reproducible results. When TLC analyzes are performed manually, they do not provide quantitative results, but only a very indicative insight into the composition of the mixture. Currently, the standard for determining the chemical composition of extracts is the use of the HPLC-MS technique. Spectrophotometric determinations are often performed as preliminary tests, e.g. determination of phenolic compounds using the Folin-Ciocalteu method, of course using appropriate standards. In turn, terpenoids are identified using GC-MS with the Kovats index, similarly to fats. It is also important to check whether isolated strains (especially Aspergillus sp.) do not produce mycotoxins.

Response: Thank you for your insightful comments. We fully agree with your assessment. In this work, our primary focus was on the isolation, characterization, and identification of five new endophytic fungi. Additionally, we highlighted the potent antibacterial activity of these fungi, with minimum inhibitory concentration (MIC) values ranging from 62.5 to 2,000 μg/mL against Escherichia coli, Staphylococcus aureus, and Enterococcus faecalis (both Gram-positive and Gram-negative bacteria). Specifically, Penicillium species showed promise as new valuable sources of secondary metabolites with significant antibacterial activities, suggesting wide-ranging applications.

We acknowledge that the chemical analysis of the composition of extracts performed in this study, primarily using manual TLC, is limited by modern standards. While our manual TLC analysis provided indicative insights, we ensured consistency by using the same amount of extract (40 µg) for comparative purposes, as mentioned in the experimental section (Section 2.4). The main objective of the TLC chromatographic analysis was to know the compound profile of these extracts so that in future work we could identify the active zones and use bioautography for bioguided isolation of the compounds of interest (Section 3.3 and 3.5).

Given the high interest and promising results, we plan to advance this project by employing more rigorous and standardized techniques. Future work will include the purification and structural elucidation of metabolites using 1D- and 2D-NMR, GC-MS (or LC-MS), to not only identify antibacterial active metabolites but also other compounds indicated in the TLC analysis. We will also re-assess the MIC of purified terpenoid in its pure form. Furthermore, we recognize the importance of ensuring the safety and finding other application of these metabolites. Thus, additional tests will be conducted to check for mycotoxins and other bioactivities such as antioxidants, cytotoxicity, and anti-diabetic effects. These expanded analyses and future perspectives are addressed in the Conclusion section. Thank you for guiding us towards a more comprehensive and rigorous approach.

Comment: The text of the publication lacks information why the breeding process was carried out for 15 days and no longer or shorter. It would be advisable to include a growth curve in the publication.

Responses: Thank you for your valuable feedback. We agree with your observation. The breeding process was performed for 15 days to ensure a standardized comparison of metabolite production across the five endophytic fungi, given their varying growth rates.

The choice of a 15-day period allowed us to observe and compare the production of secondary metabolites under consistent conditions. Also, in preliminary works in cultures of 9, 12, 15, 18 and 30 days for the endophytic fungi., similar metabolite profiles in TLC were obtained after 12 days (Section 3.3).

We recognize the need for further studies to determine the optimal conditions for active compound production, as factors such as time, temperature, light-dark exposure, and medium composition can significantly influence metabolite synthesis. Including a growth curve in future publications would provide a clearer understanding of the growth dynamics and metabolite production over time to obtain compounds of biological interest.

We have addressed these future research directions and the need for more detailed optimization studies in the Conclusion section.

Thank you for your constructive suggestion, which will help us improve the rigor and comprehensiveness of our work.

Comment: If the mycelium and the medium were extracted separately, I do not understand why these extracts were then combined together. A separate chemical analysis of the medium could reveal metabolites secreted into it. If such compounds were observed, it would significantly increase the value of the publication, because it would create opportunities to develop a continuous process in a bioreactor.

Response: Thank you for your insightful comment. We combined the mycelium and medium extracts to conduct the MIC tests and facilitate a direct comparison of antibacterial activity across the five endophytic fungi, since we observed that the active metabolite (a terpene compound) was present in both the mycelium and the medium, in especially Penicillium species. Approximately 1-2 to 4 ratio (medium to mycelium) of extract amount were obtained, while the chromatographic profile of the extracts of the medium and the mycelium were very similar (Figure R1). This was explained in the results (Section 3.5) and Discussion.

Figure R1. TLC analyzes carried out on samples SMB-24, 25 and 26 of the culture medium (left) and mycelium (right) of Penicillium species (not included in manuscript).

As you mentioned, we recognize the importance of analyzing the medium separately to identify metabolites secreted into it. This finding suggests the potential for developing a continuous process in a bioreactor, where metabolites can be harvested directly from the medium. This approach aligns with our previous response regarding the optimization of reaction conditions. These efforts will significantly increase the value of our research by providing opportunities for continuous bioprocess development. We have outlined these future research directions and potential applications in the Conclusion section.

Comment: Line 596 It should be „Gram-positive and Gram-negative” instead of „gram-positive and gram-negative”, because it came from the name of the method inventor: Hans Christian Gram, who developed the technique in 1884.

Response: Thank you for pointing this out. We have corrected it according to your comment.

Reviewer 2 Report

Comments and Suggestions for Authors

The manuscript titled “Identification, Characterization and Antibacterial Evaluation of Five Endophytic Fungi from Psychotria poeppigiana Müll. Arg. an Amazon Plant “ is devoted to study of five novel endophytic fungi species isolated from the leaves of Psychotria poeppigiana Müll. Arg, a plant from the Rubiaceae family, collected in the tropical Amazon region of Bolivia.

The endophytic fungi were identified as members of genera Neopestalotiopsis,  Penicillium, and Aspergillus sp. through 18S ribosomal RNA and ITS sequencing.  

Chemical profiling revealed that their extracts obtained by ethyl acetate contained terpenes, flavonoids, and phenolic compounds. In a bioautography study, the terpenes showed high antimicrobial activity against Escherichia coli. The extracts from the three Penicillium species exhibited potent antibacterial activity, with minimum inhibitory concentration (MIC) values ranging from 62.5 to 2,000 μg/mL against all three pathogens: Escherichia coli, Staphylococcus aureus, and Enterococcus faecalis (both Gram-positive and Gram-negative bacteria).

These endophytic fungi, especially Penicillium species could be new valuable sources of secondary metabolites with significant antibacterial activities, suggesting promising wide-range applications.

Despite the high potential interest to the subject of this manuscript, it needs a lot of additional work before publication.

1)     There are many details missing from the text, so, a careful reading and correction by authors is essential.

-        Line 102 The culture medium for the bacterial strains, Nutrient medium, and Brain Heart Infusion BHI, were purchased from MBcell, S. Korea. Commercially procured Muller-Hinton Agar, Trypto-Casein Soy Broth (TSB), and Trypto-Casein Soy Agar (TSA) were sourced from OXOID manufacturing. – Original references for applied media must be provided.  

-        For some reasons Authors included into the References instead of original references educational or industrial sites citing media and components, which are secondary sources of information, and are not scientific publications:

-        Line 707 Biology, S. Available online: https://sharebiology.com/potato-dextrose-agar-pda/ (accessed on 05, October, 2023).

-        Line 742. Research, Z. Quick-DNA™ Fungal/Bacterial Miniprep Kit. Available online: https://files.zymoresearch.com/protocols/_d6005_quick-dna_fungal-bacterial_miniprep_kit.pdf 743 (accessed on 20,January, 2024).

-        Line 745  Scientific, F. Biotium GelGreen Nucleic Acid Gel Stain, 10,000X in Water, 0.5ml. Available online: https://www.fishersci.com/shop/products/gel-green-stain-5ml/NC9728313 (accessed on 10, 746 Februery, 2024). 747

-        Line 748. Scientific, T. GeneJET PCR Purification Kit #K0701, #K0702. Available online: https://static.igem.org/mediawiki/2017/8/81/T--Chalmers-Gothenburg-- 749

-        Line 750 Thermo_Scientific_GeneJET_PCR_Purification_Kit.pdf (accessed on 12, March, 2024).

-        Line 763 EUCAST. European Committee on Antimicrobial Susceptibility Testing. Available online: https://www.eucast.org/ (accessed on 02 February 2024).

-        Line 160 Stained macroscopic glass slides using xx dye (company, country) were observed under an optical microscope (LRI - OLYMPUS-100×/0.65, Japan) [10,30], and the obtained results were compared with taxonomic keys [29,31,32]. – Please, add correct names xx dye (company, country).

-        Line 163 Furthermore, molecular taxonomy was employed for identification using specific PCR primers to amplify the endophyte 18S rDNA region and for partial sequencing of the gene encoding 18s ribosomal DNA. – Please, be more precise in description of the target sequences. So far, Authors said that they have amplified 18S rDNA and 18S rDNA.

-        Line 170 The amplification of the 18S rDNA gene region , was conducted via polymerase chain reaction (PCR) using the primers listed in Tabl1 (Table 1?) to amplify a ~530 bp fragment of the conserved region of 18s rDNA. – Please, name the positions of 18S rRNA gene and ITS amplified by the primers.

-        Line 180 The PCR products were visualized using agarose gel electrophoresis by running 2 μl of the PCR product in 1.0 % (w/v) agarose gel with 19 tris/borate electrophoresis buffer (TBE).  – What is the meaning of 19 in front of TBE?

-        Line 182 The gel was stained with GelGreen Nucleic Acid Gel Strain, 10,000× in Water (Fisher Scientific, USA), and compared with a reference marker, GeneRuler 1 Kb DNA Ladder [36].

-        Line 184 The PCR products were purified using PCR purification kit following the manufacturer protocol [37] were then subjected to direct sequencing using the same PCR primers, which were performed by Eurofins Scientific (Germany). – As stated above, manufacturer protocol may not be included into the References list.

2)     For all Figures with phylogenetic trees:

-        Please, correct all legends. The phylogenetic trees were constructed based on the target gene (18S rDNA, ITS)  analysis. Add information about the number of analyzed accessions, the analyzed gene fragment length, and number of informative nucleotides.

-        Please, remove from the trees all accessions without exact taxonomic position, unculturable samples, etc. Please, remove from the GeneBank accessions names gene attributes (they must be indicated in the Figures legends).  

-        Please, re-draw the trees to cut off all branches and remove all bootstrap values below 50%, so as most of the branches are not statistically reliable.

3)     Line 281 Figure 2. Phylogeny of the endophytic fungus Neopestalotiopsis sp. SMB-23 isolated from P. poeppigiana Müll. Arg. using EF4f/Fung5r. – It must be stated in the Figure 2 legend that SMB-23 was identified as Neopestalotiopsis sp. based on its morphological traits.

Comments on the Quality of English Language

The manuscript must be carefullly checked. There are many errors and mistyping. 

Author Response

Response to reviewers

Manuscript ID: microorganisms-3100339

Title: "Identification, Characterization and Antibacterial Evaluation of Five Endophytic Fungi from Psychotria poeppigiana Müll. Arg. an Amazon Plant"

Reviewer 2:

Comments: The manuscript titled “Identification, Characterization and Antibacterial Evaluation of Five Endophytic Fungi from Psychotria poeppigiana Müll. Arg. an Amazon Plant “is devoted to study of five novel endophytic fungi species isolated from the leaves of Psychotria poeppigiana Müll. Arg, a plant from the Rubiaceae family, collected in the tropical Amazon region of Bolivia. The endophytic fungi were identified as members of genera Neopestalotiopsis,  Penicillium, and Aspergillus sp. through 18S ribosomal RNA and ITS sequencing.  Chemical profiling revealed that their extracts obtained by ethyl acetate contained terpenes, flavonoids, and phenolic compounds. In a bioautography study, the terpenes showed high antimicrobial activity against Escherichia coli. The extracts from the three Penicillium species exhibited potent antibacterial activity, with minimum inhibitory concentration (MIC) values ranging from 62.5 to 2,000 μg/mL against all three pathogens: Escherichia coliStaphylococcus aureus, and Enterococcus faecalis (both Gram-positive and Gram-negative bacteria). These endophytic fungi, especially Penicillium species could be new valuable sources of secondary metabolites with significant antibacterial activities, suggesting promising wide-range applications. Despite the high potential interest to the subject of this manuscript, it needs a lot of additional work before publication.

Response: Thank you very much for your supportive review comments on our manuscript. We appreciate your valuable feedback and have made the necessary corrections to improve the manuscript. Please find the detailed responses below and the corresponding revisions highlighted in the re-submitted manuscript. We agree with all your comments, which are valuable for our future work on this project.

Comments: 1)     There are many details missing from the text, so, a careful reading and correction by authors is essential.

-        Line 102 The culture medium for the bacterial strains, Nutrient medium, and Brain Heart Infusion BHI, were purchased from MBcell, S. Korea. Commercially procured Muller-Hinton Agar, Trypto-Casein Soy Broth (TSB), and Trypto-Casein Soy Agar (TSA) were sourced from OXOID manufacturing. – Original references for applied media must be provided.  

Response: A reference for the preparation of Potato Dextrose Agar (PDA) and Potato Dextrose Broth (PDB) was included since these media were initially prepared in our study. However, most of the experiments were conducted using commercially available PDA and PDB. Other media used in our experiments were also commercial products purchased from reputable suppliers. Given that these are standard products widely utilized in microbiological research, we believe it is acceptable to mention the commercial sources in our methodology section. This approach ensures reproducibility and transparency in our experimental procedures.

-        For some reasons Authors included into the References instead of original references educational or industrial sites citing media and components, which are secondary sources of information, and are not scientific publications:

Response: Thank you for your observation. We have removed all manufacturer protocols (links) from the reference list, as they are typically considered standard methods for these types of experiments using the product. These protocols are widely recognized and used within the scientific community, and their inclusion in the reference list can be not necessary.

-        Line 707 Biology, S. Available online: https://sharebiology.com/potato-dextrose-agar-pda/ (accessed on 05, October, 2023).

Response: It (link) has been removed from the reference list.

-        Line 742. Research, Z. Quick-DNA™ Fungal/Bacterial Miniprep Kit. Available online: https://files.zymoresearch.com/protocols/_d6005_quick-dna_fungal-bacterial_miniprep_kit.pdf 743 (accessed on 20,January, 2024).

Response: It (link) has been removed from the reference list.

-        Line 745  Scientific, F. Biotium GelGreen Nucleic Acid Gel Stain, 10,000X in Water, 0.5ml. Available online: https://www.fishersci.com/shop/products/gel-green-stain-5ml/NC9728313 (accessed on 10, 746 Februery, 2024). 747

Response: It (link) has been removed from the reference list.

-        Line 748. Scientific, T. GeneJET PCR Purification Kit #K0701, #K0702. Available online: https://static.igem.org/mediawiki/2017/8/81/T--Chalmers-Gothenburg-- 749

Response: It (link) has been removed from the reference list.

-        Line 750 Thermo_Scientific_GeneJETe_PCR_Purification_Kit.pdf (accessed on 12, March, 2024).

Response: It (link) has been removed from the reference list.

-        Line 763 EUCAST. European Committee on Antimicrobial Susceptibility Testing. Available online: https://www.eucast.org/ (accessed on 02 February 2024).

 Response: It (link) has been removed from the reference list.

-        Line 160 Stained macroscopic glass slides using xx dye (company, country) were observed under an optical microscope (LRI - OLYMPUS-100×/0.65, Japan) [10,30], and the obtained results were compared with taxonomic keys [29,31,32]. – Please, add correct names xx dye (company, country).

Response: It has been added to the manuscript and highlighted in Red.

-        Line 163 Furthermore, molecular taxonomy was employed for identification using specific PCR primers to amplify the endophyte 18S rDNA region and for partial sequencing of the gene encoding 18s ribosomal DNA. – Please, be more precise in description of the target sequences. So far, Authors said that they have amplified 18S rDNA and 18S rDNA.

Response: It has been specified and highlighted in Red.

-        Line 170 The amplification of the 18S rDNA gene region , was conducted via polymerase chain reaction (PCR) using the primers listed in Tabl1 (Table 1?) to amplify a ~530 bp fragment of the conserved region of 18s rDNA. – Please, name the positions of 18S rRNA gene and ITS amplified by the primers.

Response: It has been added and highlighted in Red.

-        Line 180 The PCR products were visualized using agarose gel electrophoresis by running 2 μl of the PCR product in 1.0 % (w/v) agarose gel with 19 tris/borate electrophoresis buffer (TBE).  – What is the meaning of 19 in front of TBE?

Response: `19´ has been removed from the sentence.

-        Line 182 The gel was stained with GelGreen Nucleic Acid Gel Strain, 10,000× in Water (Fisher Scientific, USA), and compared with a reference marker, GeneRuler 1 Kb DNA Ladder [36].

Response: The reference of protocol (link) has been removed from reference list.

-        Line 184 The PCR products were purified using PCR purification kit following the manufacturer protocol [37] were then subjected to direct sequencing using the same PCR primers, which were performed by Eurofins Scientific (Germany). – As stated above, manufacturer protocol may not be included into the References list.

Response: The reference of protocol (link) has been removed from reference list.

Comment: 2)     For all Figures with phylogenetic trees:

-        Please, correct all legends. The phylogenetic trees were constructed based on the target gene (18S rDNA, ITS)  analysis. Add information about the number of analyzed accessions, the analyzed gene fragment length, and number of informative nucleotides.

Response: All legends were updated based on the reviewer’s comment and highlighted with red color on the main manuscript

-        Please, remove from the trees all accessions without exact taxonomic position, unculturable samples, etc. Please, remove from the GeneBank accessions names gene attributes (they must be indicated in the Figures legends).  

Response: It has been corrected accordingly.

-        Please, re-draw the trees to cut off all branches and remove all bootstrap values below 50%, so as most of the branches are not statistically reliable.

Response: The figures were re-drawn, and most of bootstrap values below 50% have been removed.

Comment: 3)     Line 281 Figure 2. Phylogeny of the endophytic fungus Neopestalotiopsis sp. SMB-23 isolated from P. poeppigiana Müll. Arg. using EF4f/Fung5r. – It must be stated in the Figure 2 legend that SMB-23 was identified as Neopestalotiopsis sp. based on its morphological traits.

Response: It has been stated in the legend of Figure 2 and highlighted in Red.

Round 2

Reviewer 1 Report

Comments and Suggestions for Authors

The authors revised the manuscript and provided extensive explanations. However, I still maintain my opinion that a journal with such a high Impact Factor cannot publish a paper without an in-depth chemical analysis. For this reason, I still believe that without an analysis of at least the extract of Penicillium sp. SMB-26 carried out using HPLC-MS (flavonoids and phenolic compounds) and GC-MS (terpenes), the work cannot be published. I would also like to point out that ethyl acetate is such a non-selective solvent that even if there is one spot on the TLC plate, it does not necessarily mean the presence of one compound. Additionally, even if it is indeed one compound, it does not necessarily have to be responsible for the biological activity of the extract, especially since in the case of extracts, a synergistic effect of their components is often observed.

In Table 4, it should be “S. aureus” instead of “S. Aureus”.

Author Response

Reviewer 1

Comment: The authors revised the manuscript and provided extensive explanations. However, I still maintain my opinion that a journal with such a high Impact Factor cannot publish a paper without an in-depth chemical analysis. For this reason, I still believe that without an analysis of at least the extract of Penicillium sp. SMB-26 carried out using HPLC-MS (flavonoids and phenolic compounds) and GC-MS (terpenes), the work cannot be published. 

Responses: Thank you for your valuable feedback. We have added UPLC analysis with UV and mass profiles, which indicate the presence of phenolic compounds in SMB-25 and SMB-26 (Figure 9) in Results section. Additionally, our GC-MS results suggest the presence of terpenes, phenolic compounds, and fatty acids in SMB-25, which we used as a representative sample (Figure 10) in Results section. We have incorporated the related information and discussion into the Discussion section along with references to relevant literature. These additions are highlighted in blue font in the manuscript.

Comment: I would also like to point out that ethyl acetate is such a non-selective solvent that even if there is one spot on the TLC plate, it does not necessarily mean the presence of one compound. Additionally, even if it is indeed one compound, it does not necessarily have to be responsible for the biological activity of the extract, especially since in the case of extracts, a synergistic effect of their components is often observed.

Response: Thank you for pointing this out, and we agree with your comments. We have reviewed several sentences in the Results and Discussion sections and made slight modifications to ensure clarity. We often use terms like 'proposed' or 'suggested' to reflect the tentative nature of our findings.

Revised Section: Page 21: Additionally, the study on the relationship between antibacterial activity and TLC-bioautography suggested that the active compound responsible for the observed antibacterial activity is a terpene. There is also a possibility of a synergistic effect of the metabolites, as non-selective solvent extracts often exhibit such interactions.

Comment: In Table 4, it should be “S. aureus” instead of “S. Aureus”.

Response: Thank you for pointing this out. We have corrected it according to your comment.

Reviewer 2 Report

Comments and Suggestions for Authors

Authors made good work improving the manuscript. There are a few items to be done before the publications, related to the  figures presenting phylogenetic trees. 

1)  NCBI accession names must be provided before the strain names on the figures 2-6. 

2) accession gene attributes like "18S small ribosomal subunit gene partial sequence"  must be removed from the trees 2, 3, 5-6 - they are obvious from the fugure legends. Accession names from the Fig. 4 must be placed before isolate name. Sourse of the sequences must be shown in the figure legend (some of them looks like not from NCBI)

3) Please, move in the Legends the  accession name close to the name of sequenced isolate - for instance: Phylogenetic tree based on the partial sequence of 18s rDNA of endophytic fungus Neopestalotiopsis sp. SMB-23 (Acession no. PP800334) obtained with F4f/Fung5r primers showing relationship by neighbor-joining with closely related taxa from NCBI GenBank, ...

4) Please, correct the Legend of Figure 4. Phylogenetic tree based on the sequence of ITS1-5.8S-ITS2 region...  The tree show analysis of accessions attributed as "Mitochondrial substrate/solute carrier"

Author Response

Reviewer 2

Authors made good work improving the manuscript. There are a few items to be done before the publications, related to the figures presenting phylogenetic trees. 

We appreciate your valuable feedback and have made the necessary corrections to improve the manuscript. Please find the detailed responses below and the corresponding revisions highlighted in the re-submitted manuscript.

Comment: 1)  NCBI accession names must be provided before the strain names on the figures 2-6. 

Response: The accession names have been added before the strain names in the legends of figures 2-6 in the manuscript. If the reviewer refers to the accession names of the species in the phylogenetic tree, we would like to clarify that these names have been obtained from the NCBI database as they are, and we prefer to keep the names as presented in the database.

Comment: 2) accession gene attributes like "18S small ribosomal subunit gene partial sequence"  must be removed from the trees 2, 3, 5-6 - they are obvious from the fugure legends. Accession names from the Fig. 4 must be placed before isolate name. Sourse of the sequences must be shown in the figure legend (some of them looks like not from NCBI)

Response: We agree with the reviewer regarding the presence of the accession gene with strain names in the phylogenetic trees. However, these names were obtained from the NCBI database as presented, without any modification. We believe it is beneficial for the reader to have the names as they are presented in the NCBI database.

Comment: 3) Please, move in the Legends the  accession name close to the name of sequenced isolate - for instance: Phylogenetic tree based on the partial sequence of 18s rDNA of endophytic fungus Neopestalotiopsis sp. SMB-23 (Acession no. PP800334) obtained with F4f/Fung5r primers showing relationship by neighbor-joining with closely related taxa from NCBI GenBank, ...

Response: We appreciate the reviewer’s suggestion and have revised all legends of figures 2-6 based on the reviewer's comment. All changes are highlighted in blue font.

Comment: 4) Please, correct the Legend of Figure 4. Phylogenetic tree based on the sequence of ITS1-5.8S-ITS2 region...  The tree show analysis of accessions attributed as "Mitochondrial substrate/solute carrier"

Response: Thank you for pointing this out. It has been corrected according to your comment.